# Enhanced Thermostability and Enzymatic Activity of cel6A Variants from *Thermobifida fusca* by Empirical Domain Engineering

**DOI:** 10.3390/biology9080214

**Published:** 2020-08-07

**Authors:** Imran Ali, Hafiz Muzzammel Rehman, Muhammad Usman Mirza, Muhammad Waheed Akhtar, Rehana Asghar, Muhammad Tariq, Rashid Ahmed, Fatima Tanveer, Hina Khalid, Huda Ahmed Alghamdi, Matheus Froeyen

**Affiliations:** 1Department of Biotechnology, Mirpur University of Science and Technology (MUST), Mirpur (AJK) 10250, Pakistan; rehanauaar@yahoo.com (R.A.); tariq.awan@must.edu.pk (M.T.); hashmi133@yahoo.com (R.A.); 2Institute of Biochemistry and Biotechnology, University of the Punjab, Lahore 54590, Pakistan; muzzammel.phd.ibb@pu.edu.pk; 3Department of Human Genetics and Molecular Biology, University of Health Sciences, Lahore 54590, Pakistan; 4Department of Pharmaceutical and Pharmacological Sciences, Rega Institute for Medical Research, Medicinal Chemistry, University of Leuven, B-3000 Leuven, Belgium; muhammadusman.mirza@kuleuven.be (M.U.M.); mathy.froeyen@kuleuven.be (M.F.); 5School of Biological Sciences, University of the Punjab, Lahore 54590, Pakistan; mwa.sbs@pu.edu.pk; 6Department of Biotechnology, Forman Christian College University, Lahore 54590, Pakistan; 19-10544@formanite.fccollege.edu.pk; 7Shaheed Zulfiqar Ali Bhutto Medical University, Islamabad 44000, Pakistan; hina.khalid.mehmood@gmail.com; 8Department of Biology, College of Sciences, King Khalid University, Abha 61413, Saudi Arabia; hudaghamdi@kku.edu.sa

**Keywords:** endoglucanase (cel6A), domain engineering, *Thermobifida fusca*, molecular dynamics simulations, thermostable enzymes

## Abstract

Cellulases are a set of lignocellulolytic enzymes, capable of producing eco-friendly low-cost renewable bioethanol. However, low stability and hydrolytic activity limit their wide-scale applicability at the industrial scale. In this work, we report the domain engineering of endoglucanase (cel6A) of *Thermobifida fusca* to improve their catalytic activity and thermal stability. Later, enzymatic activity and thermostability of the most efficient variant named as cel6A.CBC was analyzed by molecular dynamics simulations. This variant demonstrated profound activity against soluble and insoluble cellulosic substrates like filter paper, alkali-treated bagasse, regenerated amorphous cellulose (RAC), and bacterial microcrystalline cellulose. The variant cel6A.CBC showed the highest catalysis of carboxymethyl cellulose (CMC) and other related insoluble substrates at a pH of 6.0 and a temperature of 60 °C. Furthermore, a sound rationale was observed between experimental findings and molecular modeling of cel6A.CBC which revealed thermostability of cel6A.CBC at 26.85, 60.85, and 74.85 °C as well as structural flexibility at 126.85 °C. Therefore, a thermostable derivative of cel6A engineered in the present work has enhanced biological performance and can be a useful construct for the mass production of bioethanol from plant biomass.

## 1. Introduction

Cellulose is the most abundant plant biomass found on earth and is extremely important for mankind due to its diverse applications. The evolving use of renewable non-food cellulosic plant biomass is a very attractive option for biofuel production. Biological depolymerization of cellulose biomass is accomplished by various types of cellulases, particularly endoglucanases, exoglucanases, and β-glucosidases. Endoglucanase cleaves cellulose polymer into oligopolymers of varying lengths, while exoglucanases attack reducing and/or non-reducing ends of cellulose yielding glucose or cellobioses, finally β-glucosidases convert cellobioses into fermentable sugar monomers. Despite the tremendous potential of cellulases in the conversion of plant biomass into useable bioethanol, their widespread application is restricted by their high cost and low efficacy, especially under harsh industrial conditions [1,2,3,4,5]. Enzymes with high thermostability are advantageous in saccharification processes owing to their better penetrating ability into lignocellulosic biomass for disorganization [6]. Furthermore, plant biomass degradations at elevated temperatures reduce the risk of microbial contamination and cooling costs after biomass pretreatment [7]. Therefore, tailor-made cellulases by molecular engineering with improved catalytic activity and thermostability is a prerequisite in order to be used in industrial applications.

*Thermobifida fusca* is a moderately thermophilic, major lignocellulolytic soil bacterium. It has been well known for its potential to produce various types of cellulases that can penetrate cellulose and lignocellulose residues to yield simple saccharide units [8]. Among various glycosyl hydrolase family 6 endoglucanases, endoglucanase (cel6A) of *T. fusca* is considered to be the most efficient reducing sugar producer, hence has the potential to be used in the biofuel industry [9]. Structurally, cel6A is a modular enzyme with a non-catalytic family 2 carbohydrate-binding module (CBM-II) at C-terminal. CBMs facilitate the adsorption of catalytic domain with the substrates [10], help in the alignment of enzyme-substrate in close proximity [11,12], and carry out hydrolysis by modification of substrate surfaces [13,14].

There are very few reports on enzyme engineering for industrial applicability, particularly to improve the thermostability of cellulases [1,15,16]. Thermostable multimodular cellulases provide an excellent template for modification to enhance their suitability for industrial applications. The accessory roles of non-catalytic domains (CBMs) on thermal stability and insoluble substrate degrading efficiencies have been extensively reported by removing or grafting from respective catalytic domain/s [17,18,19,20,21,22]. MD simulations (molecular dynamics, is used to analyze physical movements of atoms and molecules by using computational tools) have been used in several studies to evaluate the factors regulating thermostability of enzymes [11,23,24,25,26,27,28,29,30] and to investigate the stability of the biomolecular complex and interaction attributes [31,32]. These include the thermal stable mechanisms of rubredoxin [33], nuclease [34], barnase [35], nitrile hydratase [36], adenylate kinase [37], carbonic anhydrase [38], carboxylesterase from *Geobacillus stearothermophilus* [39], psychrophilic esterase from *Pseudoalteromonas haloplanktis* [40], hyperthermophilic esterase from *Archaeoglobus fulgidus* [41], thermostable para-nitrobenzyl esterase from *Bacillus subtilis* [42], and CBMs from *Clostridium cellulovorans* [43,44].

In this study, we designed a group of cel6A variants, by deletion, insertion, and rearrangement of CBM-II as well as a catalytic domain (GH6). The engineered variants were expressed and characterized for different properties such as thermal stability and catalytic activity at different temperatures. Furthermore, catalytic functionality of cel6A variant was confirmed by application of multi-template homology modelling by using MODELLER v9.15. Thermostability of cel6A construct was investigated structurally to get an insight into the dynamic influence at different comparable temperatures. MD simulations were performed at four different temperatures (26.85, 60.85, 74.85, and 126.85 °C) to develop a most effective rationale for the assessment of the impact of the factors governing enzyme’s thermostability. The dynamic consequences of the enzyme construct were explored by calculating the root mean square deviation (RMSD) and root mean square fluctuation (RMSF) values for all Cα backbone atoms to identify the thermal sensitive regions as performed in some earlier studies [35,36,37,43]. The result obtained showed that cel6A is most active on soluble and insoluble cellulosic substrates. cel6A.CBC exhibited highest catalysis of carboxymethyl cellulose (CMC) at a pH 6.0 and 60 °C. Furthermore, cel6A.CBC showed a good thermostability at 26.85, 60.85, and 74.85 °C as well as structural flexibility at 126.85 °C by molecular modelling assay. Therefore, a thermostable derivative of cel6A engineered in the present work showed a better performance and can be a useful variant for the production of bioethanol from cellulose.

## 2. Materials and Methods

### 2.1. Reagents and Chemicals

Plasmid and gene purification kits were purchased from GeneAll (Seoul, Korea). Restriction enzymes, T4 DNA ligase, and DNA/Protein markers were purchased from Thermo Scientific (Mississauga, Ontario, Canada). All other chemicals and reagents used in this study were purchased from Sigma Aldrich.

### 2.2. PCR Amplification of cel6A Variants

Construction scheme of cel6A derivatives is shown in Appendix A. All derivatives were amplified from cel6A gene as template [45] using the respective primers (Appendix A). A PCR reaction mixture of 50 µL was prepared as 1X *Taq* buffer, 0.4 mM dNTPs, 1 µM concentration of forward and reverse primers, ~100 ng template DNA and 2 IU of *Taq* DNA polymerase. Gene amplification was done on Thermocycler Star 96, IRMECO with thermal conditions of 95 °C for 3 min as initial denaturation, followed by 30 cycles of 95 °C for 30 s denaturation, annealing at 58 °C for 30 s, and extension at 72 °C. Final extension of 20 min at 72 °C was also done. Amplified gene segments were gel purified and cloned into pTZ57R/T by InsT/A cloning kit. Preparation of *E. coli* DH5α competent cells and transformation was done according to the standard protocols [46].

### 2.3. Sub Cloning of cel6A Variants in Expression Vector

Recombinant plasmids [pTZ-cel6A.C, -cel6A.BC, -cel6A.BCB, -cel6A.CBC, -cel6A.CBCB, and pET22b(+)] were extracted by GeneAll plasmid extraction kit, digested with respective restriction enzymes, and purified from 0.8% agarose gel by gel extraction kit (GeneAll). Sequential cloning (Appendix A) was done to ligate different gene segments into the expression vector pET22b(+) to generate truncated, CBM-II transposition, addition, catalytic domain addition, and duplication of cel6A variants. The *E. coli* (DH5α) cells were chemically made competent according to the standard protocol [46]. These cells were transformed with cel6A variants, spread on ampicillin agar plates and incubated overnight at 37 °C. Transformants were confirmed by colony pick PCR and endonuclease digestion.

### 2.4. Expression Analysis of cel6A Variants

*E. coli* (BL21-CodonPlus (DE3)-RIPL) cells were chemically made competent and transformed with p*cel6A*.C, pcel6A.BC, pcel6A.BCB, pcel6A.CBC, pcel6A.CBCB. A single transformed colony was resuspended in 10 mL LB broth containing ampicillin (100 µg.mL^−1^) and incubated at 37 °C for 16 h. Fresh LB broth (100 mL) containing antibiotic was inoculated with 0.5 mL overnight grown transformed BL21 cells and placed at 37 °C with 250 rpm shaking in an orbital shaker. Cells were induced with 0.5 mM IPTG when OD_600_ reached 0.4–0.6 and expression was analyzed on SDS-PAGE after 8 h of induction. For partial purification of cel6A engineered enzymes, induced cells were harvested at 6500 rpm for 10 min and resuspended in 50 mM phosphate buffer (pH 6) to final OD_600_ 1.0. Cells were lysed by ultra-sonication (Sonics vcx500, newtown, Connecticut, USA). Samples were processed at 30% amplitude for 30 min with 2.5 sec pulse and 1 min interval. The lysate was heated at 60 °C for 30 min and centrifuged at 6500 rpm for 15 min. The supernatant was taken for enzymatic and physiological characterization.

### 2.5. Enzymatic Activity Assay

Endoglucanase activity of cel6A variants was measured by incubating appropriately diluted enzyme solution (0.5 mL) with 1% carboxymethyl cellulose (CMC) as soluble substrate solution in 0.05 M phosphate buffer having pH 6.0 at 60 °C for 10 min. The reaction was halted by adding 0.5 mL Na_2_CO_3_ (1M). Reducing sugars released by reaction was measured by adding 3 mL 3,5-Dinitrosalicylic acid (DNS) reagent (5.3 g of DNS, 9.9 g NaOH, 15.3 g Na-K-tartarate and 4.15 g Na metabisulphite in 708 mL of water and adding 3.8 mL of phenol). The OD was measured at 600 nm against a blank [47]. Enzyme activity (U) is the number of micromoles (µm) of reducing sugars equivalent produced per minute under experimental conditions.

For hydrolytic activity assay on insoluble substrates, pre-treated bagasse, filter paper (FP), regenerated amorphous cellulose (RAC), and bacterial microcrystalline cellulose (BMCC) were used. BMCC was prepared by adding 5 g BMCC in distilled water containing 0.02% sodium azide and stirring overnight at 4 °C. Bagasse was prepared by autoclaving (15 lb.in^−2^) grounded bagasse powder with 0.5% NaOH for 3 h. Alkali treated bagasse was washed with distilled water to neutrality and dried. Activity assay was done by taking 10 mg each of the substrates in an aliquot containing enzyme in 1 mL of 50 mM phosphate buffer (pH 6.0). Reducing sugars were measured after 12 h incubation in shaking water bath (60 °C) by DNS method. Blanks were prepared by the same procedure but without adding the enzyme. The concentration of the protein was determined using BSA (bovine serum albumin) as standard by the dye-binding method [48]. Assays were performed in triplicates. All enzymatic assays were performed as reported in previous studies [49,50].

### 2.6. pH Stability Determination

Optimum pH for cel6A variants was determined by suitably diluting enzymes with 50 mM acetate, phosphate, Tris-Cl, and borate-NaOH buffers, pH 3.0–5.0, 5.5–7.5, 8.0–9.0, and 9.5–10.0, respectively. The pH stability was determined by incubating endoglucanase variants at pH 3.0–10.0 for 120 min at room temperature (25 °C), and residual activity were assayed at regular intervals (20 min).

### 2.7. Thermal Stability Determination

Thermostability of cel6A variants was determined by incubating enzymes at different thermal conditions ranging from 50 to 75 °C for varying intervals up to 120 min and residual activity was assayed by incubating 500 µl enzyme with 500 µL of carboxymethyl cellulose (1% *w*/*v*) as substrate dissolved in 50 mM phosphate buffer (pH 6.0) and resulting reducing sugars were determined by DNS method.

### 2.8. Molecular Modeling Study

Based on results of enzymatic assays, only the best thermostable construct was examined for better insight into the structural stability of domains on different comparable temperatures, namely, 26.85, 60.85, 74.85, and 126.85 °C. For the structural study, engineered *cel6A* variants were modelled through multi-template homology modelling by using MODELLER v9.15 [51,52]. The generated models were optimized and refined using a short 20-ns molecular dynamics (MD) simulation at standard temperature (26.85 °C) and pressure (1 atm). The stereochemical assessment of modelled structures and residue-by-residue geometry were validated by Molprobity [53]. All MD simulations were carried out using AMBER 18 simulation package [54]. The same MD simulation protocol was set up using the AMBER ff99SB force field [55], TIP3 water molecule model system [56], and neutralizing ions by creating an octahedral box extended 12.0 Å around the solute. The same initial energy minimization, equilibration of the simulation system, and the following standard production run was utilized as described elsewhere [31,57,58]. To run MD simulation at different temperatures, the minimized systems of cel6A construct were equilibrated for 100 ps each at four temperatures (26.85, 60.85, 74.85, and 126.85 °C) by position restrained molecular dynamics simulation to relax the solvent. The equilibrated systems were then subjected to a production run of 100 ns each at four different temperatures. The CPPTRAJ module of AMBER 18 was utilized for the trajectory analysis. The representative model was extracted after MD simulation and analyzed through Chimera v1.13 [59]. Molecular docking was performed using AutoDock Vina [60] to analyze the binding conformation of CMC substrate in respective catalytic domain. Binding pocket information was extracted by superimposing the co-crystalized catalytic domain of endo-1,4-glucanase cel6A from *Thermobifida fusca* in complex with methyl cellobiosyl-4-thio-β- cellobioside (PDB ID: 2BOG) [61] and docking grid was exclusively build around the binding pocket. Details of protein preparation, optimization, and minimization are described in previous studies [57,62].

In order to access the quantitative description of binding affinity of cel6A constructs with bound CMC substrate, molecular mechanics-generalized born surface area (MM-GBSA) binding free energy calculations were performed using the AMBER 18 mmgbsa module. Binding free energy calculations by MMGBSA has been extensively discussed [63,64]. Total of 1000 snapshots from the whole MD trajectory of the complex was generated and binding free energy (ΔG_total_), was calculated, ΔG_total_ is the sum of molecular mechanics energy (ΔE_MM_), solvation free energy (ΔG_sol_) contributions. ΔE_MM_ is further divided into internal energy (ΔE_int_), electrostatic (ΔE_ele_), van der Waals (ΔE_vdw_) energy in the gas phase, whereas and ΔG_sol_ is divided into polar (ΔG_p_) and non-polar (ΔG_np_) contributions to the solvation free energy, as follows:$$\Delta E_{\mathrm{MM}}=\Delta E_{\mathrm{int}}+\Delta E_{\mathrm{ele}}+\Delta E_{\mathrm{vdw}}\;\Delta G_{\mathrm{sol}}=\Delta G_{\mathrm{polar}}+\Delta G_{\mathrm{nonpolar}}\;\Delta G_{\mathrm{total}}=\Delta E_{\mathrm{MM}}+\Delta G_{\mathrm{sol}}$$

## 3. Results and Discussion

Non-catalytic family 2 carbohydrate-binding module (CBM-II) and catalytic domain glycosyl hydrolase-6 of endoglucanase cel6A were either deleted, inserted, or rearranged to engineer a suitable variant of cel6A that can withstand harsh conditions of pH and temperature for industrial applications. Then engineered cel6A variants were expressed and assessed for their enzymatic activity. The cel6A construct with higher thermostability was investigated structurally through long-run molecular dynamics simulations to get an insight of the thermostable behavior of enzyme at different comparable temperatures.

### 3.1. Construction of Plasmids for Domain Engineered cel6A Variants

Endoglucanase variants were constructed from catalytic and non-catalytic domains by addition, truncation, and transposition (Appendix A). All gene fragments after PCR amplification were gel purified, TA-cloned in pTZ57R/T vector and *E. coli* DH5α cells were then transformed with recombinant vectors. The first construct encoding catalytic domain (cel6A.C) without CBM-II was sub-cloned after restriction from a pTZ-cel6A.C plasmid into T7 promoter-based pET22b(+) expression vector (Appendix A). Successful integration of the catalytic domain was confirmed by digesting pcel6A.C plasmid with NdeI and HindIII endonucleases, which released a fragment of 0.86 kb of cel6A.C.

To evaluate the positional/spatial effect of CBM-II, transposition of CBM-II from C-terminal of cel6A.CB to N-terminal was done by sequential cloning of CBM-II (272 bp), linker (110 bp), and then cel6A.Cn into pET22b(+) expression vector to generate pcel6A.BC (Appendix A). Restriction digestion of pcel6A.BC with NdeI and HindIII generated a 1.3 kb fragment corroborating the transposition of CBM-II from C-terminal to the N-terminal.

Later, three other variants; cel6A.BCB, cel6A.CBC, and cel6A.CBCB were constructed. A gene fragment was nicked from pTZ-cel6A.CBn with BamHI and HindIII with an accurate size of 1.3 kb ligated to the pCBM-II-L vector digested with same endonucleases to generate pcel6A.BCB. The successful insertion was confirmed by the presence of 1.7 kb fragment after digesting with NdeI and HindIII enzymes. Similarly, additional catalytic domains and complete gene were inserted by removing stop codon and insertion of NcoI site at C-terminal of cel6A gene to generate pcel6A.CBCB. Inserts to be added were amplified with NcoI and HindIII sites at N-terminal of the catalytic domain and cel6A construct. Both cel6A.CBC and cel6A.CBCB was digested with NdeI and HindIII endonucleases to generate 2.2 and 2.7 kb restricted fragments, respectively, which substantiated the successful insertion (Appendix A). All constructs were sequenced in order to determine their inframe insertion and sequence fidelity. Other researchers had adopted similar domain rearrangement strategies for xylanase and endoglucanase genes of *C. thermocellum* [49,50,65].

### 3.2. Expression Analysis and Enzymatic Activities of cel6A Variants

Recombinant expression of cel6A variants in *E. coli* BL21 CodonPlus (RIPL) was analysed in LB broth using IPTG as an inducer. All the variants were successfully expressed in *E. coli* BL21 CodonPlus (RIPL). Samples were taken after 8 h of 0.5 mM IPTG induction, and the expression percentage of each variant was densitometrically calculated using Gene Tools software (G-box, Syngene). Protein expression levels of cel6A.C and cel6A.CBC was higher than all other constructs, while cel6A.BCBC showed minimum expression level (Table 1). The cells were lysed to evaluate the subcellular localization of cel6A variants. All variants were expressed in the soluble cytoplasmic fraction (Figure 1).

The enzymatic activity on soluble substrate CMC for cel6A.C, cel6A.BC, cel6A.BCB, cel6A.CBC, and cel6A.CBCB were 320, 290, 170, 600, and 250 U l^−1^ OD600^−1^, respectively, while specific activities of these variants were 4.2, 4.3, 3.0, 7.2, and 6.9 U mg^−1^ enzymes, respectively (Table 1). Irwin et al., 1993 [66] carried out enzymatic activities of native cel6A and cel6A catalytic domain on different soluble and insoluble substrates, which supports our results of native cel6A.C and Cel6.BC. In a similar study, a fusion of CBM-II from Cel6B of *T. fusca* to the C- terminal of Cel5A of *T. maritima* and Cel9A of *Alicyclobacillus acidocaldarius* catalytic domains resulted in the improved hydrolytic activity of these engineered enzymes as compared to the native enzymes [28]. Our results showed enhanced endoglucanase activity by domain variant cel6A.CBC in comparison with native cel6A.C and cel6A.BC. Catalysis enhancement of the catalytic domains fused with CBMs on insoluble substrates has been reported previously due to many factors, including enzyme-substrate proximity [10]. Hydrolytic effect of cel6A variants on insoluble substrates was more pronounced. cel6A.C, cel6A.BC, cel6A.BCB, cel6ACBC, and cel6A.CBCB liberated 0.7, 1.0, 1.7, 2.0, and 1.5 μM reducing sugars per μM^−1^ enzyme, respectively, from filter paper (FP). A similar pattern of hydrolysis was observed on other insoluble substrates as well. cel6A.BCB and cel6A.CBC was found more active on all insoluble substrates. cel6A.BCB produced 1.4, 2.6, and 0.28 μM, reducing sugars per μM^−1^ enzyme and cel6A.CBC produced 3.4, 5.4, and 0.9 μM, reducing sugars per μM^−1^ enzyme from bagasse, RAC, and BMCC, respectively (Table 1). Enzymatic activities of native cel6A were almost similar to that of cel6A.BC construct on soluble and insoluble substrates, which we reported previously [49,50], Similar results were reported in a study in which an extra catalytic domain was added to the CelA of *C. thermocellum* [65]. In another study, the addition of CBMs to different endoglucanases increased their hydrolytic activities on Avicel [16].

### 3.3. Physiological Characterization of cel6A Variants

#### 3.3.1. Effect of Temperature on cel6A Variants

The endoglucanase activity of engineered cel6A constructs showed almost similar hydrolytic activity pattern by retaining more than 80% endoglucanase activity at 60–70 °C. The optimum temperature for endoglucanase activity was 60 °C for all the cel6A variants where they showed 100% enzymatic activity. There was a significant difference between the enzymatic activities of native cel6A and engineered domain variants. The variant cel6A.CBC showed the most thermophilic property. The cel6A.CBC depicted enhanced endoglucanase activity in comparison with native cel6A at 70 (87% vs. 84%), at 75 (73% vs. 67%), and at 80 (45% vs. 33%). (Figure 2).

Thermostability assays of the recombinant constructs were performed at six pre-incubation temperatures (50, 55, 60, 65, 70, and 75 °C), for 2 h. At 50, 55, and 60 °C pre-incubation temperatures, all recombinant enzymes as well as native enzyme maintained more than 85% of residual activity after 2 h of incubation showing no appreciable difference in their activity. On the other hand, in a reaction assay at a pre-incubation temperature of 65 °C, a slight difference was observed in the residual activity of different recombinant constructs. The observed residual activities after 2 h incubation were 64, 71, 64, 58, 78, and 61 of recombinant enzymes cel6A.CB (cel6A.CB is native cel6A), cel6A.C, cel6A.BC, cel6A.BCB, cel6A.CBC, and cel6A.CBCB, respectively. There was a substantial difference between the residual activities of cel6A.CBC and other enzymes at pre-incubation temperatures of 70 and 75 °C. The cel6A.CBC maintained a residual activity of 62% at 70 °C after 60 min while all other enzymes decreased to less than 30%. On pre-incubation at 75 °C, cel6A.CBC and cel6A.C showed residual activity of 24% and 20%, respectively, after 60 min and 10% and 0% after 90 min, whereas all other enzymes lost catalytic activity at 75 °C in less than 60 min (Figure 3).

#### 3.3.2. Effect of pH on cel6A Variants

Optimum enzymatic activity of endoglucanase variants was measured on a broad pH range of 3.0 to 10.0. All variants retained more than 80% hydrolytic activity between pH 5.0–8.0. The hydrolytic assay was performed after incubating each of cel6A variants at any of this pH at room temperature for 2 h. Enzymatic activity gradually decreased below pH 5.0 and above pH 8.0, with optimum activity at pH 6.5 for all cel6A variants (Figure 4). cel6A. cel6A.CBC retained more enzymatic activity in comparison with native cel6A at pH 9.0 (66% vs. 61%), at pH 9.5 (58% vs. 55%), and at pH 10.0 (53% vs. 51%).

### 3.4. Molecular Modeling Analysis

Among all thermostable engineered cel6A constructs, cel6A.CBC showed enhanced enzymatic activity and was found to be more stable at a higher temperature (Figure 2 and Figure 3). It was interesting to explore the temperature influence on the dynamics of cel6A.CBC. Homology model of cel6A. CBC construct was generated by a multi-template approach using MODELLER. To build cel6A.CBC model, an X-ray resolved endoglucanase structure from *Thermobifida fusca* was retrieved (PDB ID: 2BOG) [61] to generate N (residues 1–287) and C-catalytic domain (residues 407–692), whereas a carbohydrate-binding type-2 (CBM-II) domain (PDB ID: 3NDZ; Identity: 42%; Probability: 98%; E-value: 2.3 × 10^−5^) was used to build the binding domain (residues 288–406). The wild-type cel6A.CB model was merely generated by truncating the C-catalytic domain from cel6A.CBC model. Both models were optimized and refined using a short 20 ns MD simulation. Then, these models were validated using MolProbity to check all-atom contacts and geometry (Table 2). The MolProbity score, which combines the clash score, rotamer, and Ramachandran estimations into a single score (normalized to be on the same scale as X-ray resolution), was 1.68 and 1.70 (99th percentile) for cel6A.CBC and cel6A.CB respectively. The MD optimized cel6A.CBC model showed 90.27% (622/689) residues, while cel6A.CB exhibited 90.9% (370/407) residues in Ramachandran favoured (>98%) regions. Furthermore, cel6A.CBC model showed 97.9% (674/689) residues placed in Ramachandran allowed region (>99.8%) with 11 (1.59%) outliers, whereas cel6A.CB model showed 96.8% (394/407) residues with 7 outliers (1.71%) (Table 2). After model generation, molecular docking studies were also carried out to explore the protein-ligand associations. Due to the presence of the same catalytic unit in both models having the same binding site residues, the docking affinities of CMC were similar (docking score: −6.9 Kcal/mol) in both constructs.

### 3.5. Global Structural Stability

Since protein denaturation usually arises in microsecond time scale [67,68], it is difficult to investigate the protein unfolding at normal temperatures using molecular dynamic simulations. To analyze denaturation process in cel6A constructs within the reasonable time limits, much higher temperatures are used. MD simulations procedures have been performed at higher temperatures previously to study the thermostability in various enzymes [15,36,38,69,70]. To investigate the global structural stability of cel6A.CBC, a comparative molecular dynamics (MD) simulation, was carried out at different temperatures. Based on the endoglucanase activity results, four temperatures were selected, including room temperature (26.85 °C), the optimal temperature for endoglucanase activity by all constructs (60.85 °C), the highest temperature where cel6A.CBC still showed slight thermostability with more than 80% CMCase activity (74.85 °C), and one extreme temperature (126.85 °C; to explore the thermal sensitive regions). MD simulations for a total of 100 ns were carried out at each temperature range to observe deviations and fluctuations implicated in conformational changes of cel6A.CBC.

Structural thermal fluctuations of proteins are intrinsically related to their functions [71]. Therefore, we initially investigated the temperature influence on global structure stability in terms of root mean square deviations (RMSD) of the cel6A.CBC Cα-backbone atoms during the simulation. All MD simulations analysis is illustrated in Figure 5. Figure 5A evaluates the backbone RMSD trajectories from the corresponding initial structure as a function of time at four different temperatures. As expected, cel6A.CBC remained stable during the entire simulation period of 100 ns at 26.85 °C and 60.85 °C, and converged between 1 to 1.5Å and showed close resemblance with the initial structures (Figure 5A and Figure 6). Throughout the simulation, both catalytic domains pulled inward over the CBM-II domain and adopted a closed conformation over time (Figure 6A–C). At 74.85 °C, where cel6A.CBC showed CMCase activity, and all other constructs lost their activities, the cel6A.CBC Cα-backbone atoms remained stable for first 20 ns and reached equilibrium at about 4Å and attained a value of 5Å which showed stable RMSD for the rest of the simulation. Although it fluctuated slightly in that period but remained converged within 1.5Å radius, demonstrated more favourable conformation. At 126.85 °C (400 K) simulations, cel6A.CBC Cα-backbone atoms increased in the beginning, slightly stable at 4Å, and attained a high value of 7.5 Å (Figure 5A). Thus, increasing temperature up to 126.85 °C (400 K) showed little stability and significant structural distortions were examined only at a higher temperature of 126.85 °C.

### 3.6. Structural Flexibility

To confirm the stability of cel6A.CBC at different temperatures, the average root-mean-square-fluctuations (RMSF) were also calculated, which could give us the qualitative comparison of the fluctuations of different regions of cel6A.CBC construct. Figure 5B compares the RMSFs of each residue for every temperature simulations for cel6A.CBC. At 26.85 and 60.85 °C, most regions of cel6A.CBC showed slight fluctuations at increasing temperature, indicated cel6A.CBC is rather thermostable at these temperatures. At 74.85 °C, although the N-terminal catalytic domain showed less fluctuations, CBM-II domain connecting to the C-terminal catalytic domain experienced significant fluctuations, which directed larger fluctuations up to ~4Å in C-terminal catalytic domain. These results agreed with the experimental thermostability assays, where cel6A.CBC still showed slight thermostability at 74.85 °C with more than 80% CMCase activity. When the temperature was elevated to 126.85 °C, most of the residues became highly mobile due to the denaturing of secondary structural at extreme temperature. Whereas, the linker displayed systematically less fluctuations throughout simulation at other temperatures (Figure 5B). Figure 6D clearly illustrated the loss of β-sheet content in CBM-II domain, which initiated the unfolding process and both catalytic domains moved apart at a higher temperature (at 126.85 °C). Contrary to that, both catalytic domains remained closer in a compact confirmation at other temperatures (26.85, 60.85, and 74.85 °C) throughout 100-ns simulations (Figure 5 and Figure 6).

### 3.7. Binding Free Energy Calculations

In order to provide quantitative descriptions, i.e., the absolute binding free energies of cel6A.CBC to the CMC substrate binding, MM-GBSA module of Amber 18 was utilized to analyze the intermolecular contributions with bound CMC. In order to compare the binding free energy, cel6A.CBC was compared to its wild-type cel6A.CB., as cel6A.CBC contained two catalytic domains. Therefore a docked complex was obtained bound with two CMC molecules in both catalytic domains, as shown in Figure 7.

For wild-type cel6A.CBC, a simulation system was configured with both CMC molecules, and MD simulation was performed for 20 ns. For wild-type cel6A.CB, a best-docked conformation bound with CMC molecule was processed for the same time duration. A total of 1000 snapshots were obtained from the whole 20 ns MD trajectory of each complex, and average binding free energy (ΔG_total_) was calculated (Table 3). Although there was no direct comparison for the MM-GBSA results obtained from this study, the reliability of MM-GBSA method in the calculation of substrate binding free energies has been descriptively studied for the recognition of cellulose by the CBM [43,44]. The calculated binding free energy of cel6A.CBC/CMC complex (ΔG_total_ = −20.6 kcal/mol) was lower than cel6A.CB/CMC (ΔG_total_ = −12.0 kcal/mol), suggested favorable binding affinity in comparison. Such a difference in binding free energies was evident due to the presence of an additional catalytic domain in cel6A.CBC, which established additional intermolecular interactions with the second CMC molecule. These results were in fairly good agreement with the experiment, where higher endoglucanase activity (600 U l^−1^ OD 600^−1^) and specific CMCase activity (7.2 U mg^−1^ enzyme) were observed for cel6A.CBC construct as compared to the other constructs (Table 1). MM-GBSA values of cel6A.CBC/CMC complex was also observed at different temperatures (using the same procedure as described above) to further insight into the binding energy differences. The MM-GBSA values remained consistent for the first three temperatures [ΔG_total_ = −20.6 kcal/mol at 26.85 °C (300 K); −21.5 kcal/mol at 60.85 °C (334 K); −19.2 kcal/mol at 74.85 °C (348 K)] but gradually reduced to −8.9 kcal/mol at 126.85 °C (Table 3). The energy difference at the extreme 126.85 °C temperature was obvious from the dramatic distortion of C-terminal catalytic domain due to the loss of β-sheet content in the CBM-II domain (Figure 6D), which triggered higher fluctuations in the catalytic domain bound to CMC substrate, thus diminished catalytically important electrostatic and van der Waals interactions.

## 4. Conclusions

All cel6A variants showed a broad range of pH optima with a pH range of 5.0 and 8.0 retaining more than 80% activity, except cel6A.CBC, which retained more than 90% endoglucanase activity in this range. The optimum pH for all of the cel6A variants was 6.5. All the cel6A variants showed more than 80% endoglucanase activity at a temperature range of 55 and 70 °C with optimum CMCase activity at 65 °C. cel6A.C without CBM and cel6A.CBC with a new catalytic domain showed slightly higher thermostability with more than 80% CMCase activity at 75 °C after 15 min of incubation and more than 60% activity after 30 min incubation. These results suggested that inclusion of an extra catalytic domain plays an important role in the thermostability of enzyme, while additional CBM reduces its thermostability. Moreover, the MD simulations study showed an excellent rationale with the experimental results and emphasized the thermostable region at higher temperatures. The present study focused on the thermostable cel6A construct, which can potentiate the mass production of bioethanol from plant biomass. On the other side, this study also identified the factors responsible for thermostability of endoglucanase (cel6A) variants from *Thermobifida fusca* that may endeavour to design enzymes with enhanced thermostability.

## Figures and Tables

**Figure 1 biology-09-00214-f001:**
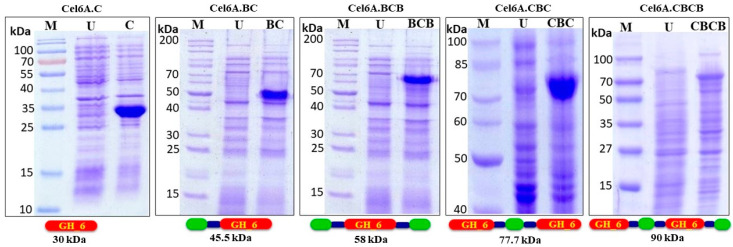
SDS-PAGE (12%) analysis of total *E. coli* BL-21 CodonPlus (RIPL) proteins expressing cel6A.C, cel6A.BC, cel6A.BCB, cel6A.CBC and cel6A.CBCB after induction with 0.4 mM IPTG. **M**: Protein Marker, **U**: Uninduced sample.

**Figure 2 biology-09-00214-f002:**
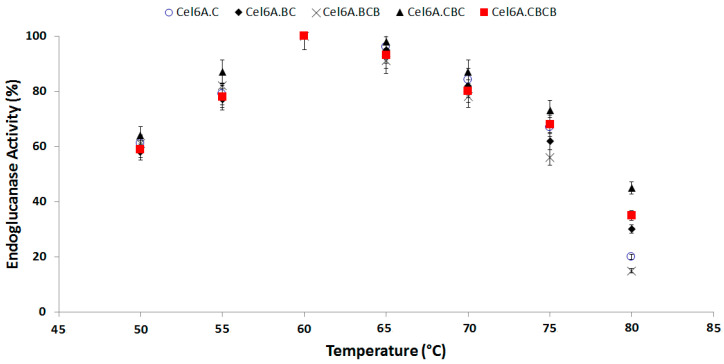
Effect of Temperature on hydrolytic activity of cel6A variants against CMC.

**Figure 3 biology-09-00214-f003:**
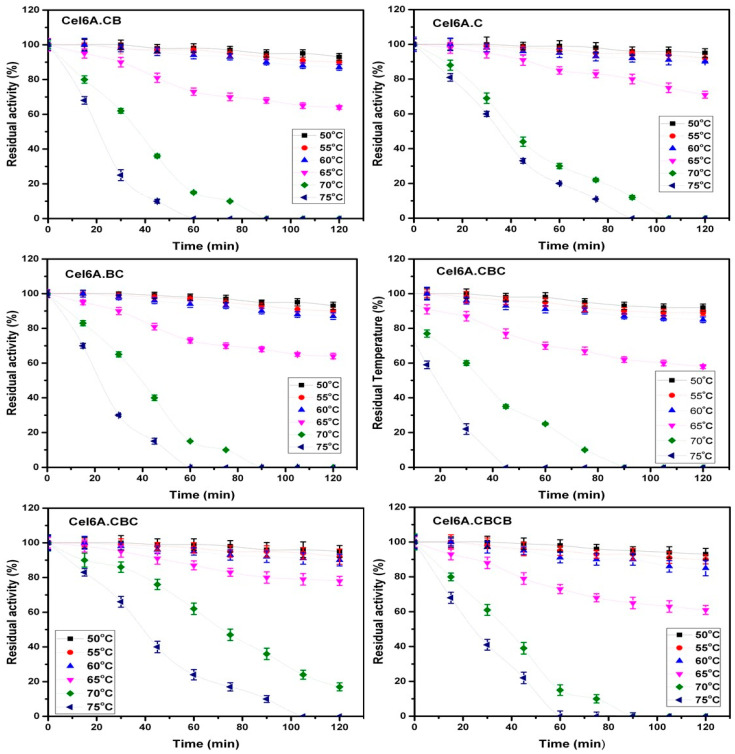
Residual activities of cel6A variants after incubation at varying temperatures.

**Figure 4 biology-09-00214-f004:**
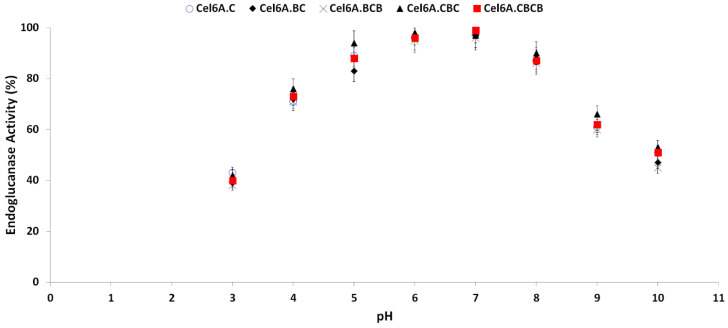
Effect of pH on hydrolytic activity against CMC of cel6A variants.

**Figure 5 biology-09-00214-f005:**
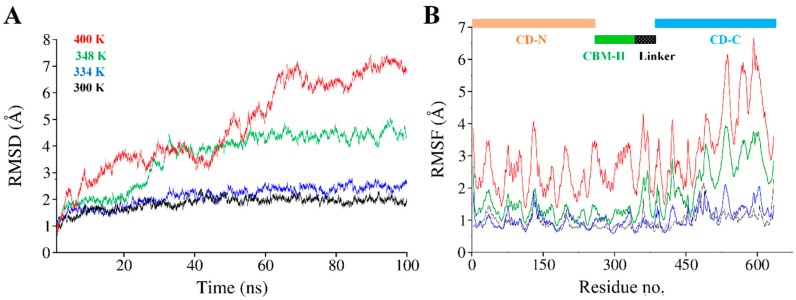
(**A**) Time dependent all backbone atom root mean square deviation (RMSD) of cel6A.CBC, (**B**) Root mean square fluctuation (RMSF) as a function of residue number of cel6A.CBC at different temperatures. Each trajectory is colored as follows: 26.85 °C (black), 60.85 °C (blue), 74.85 °C (green), and 126.85 °C (red).

**Figure 6 biology-09-00214-f006:**
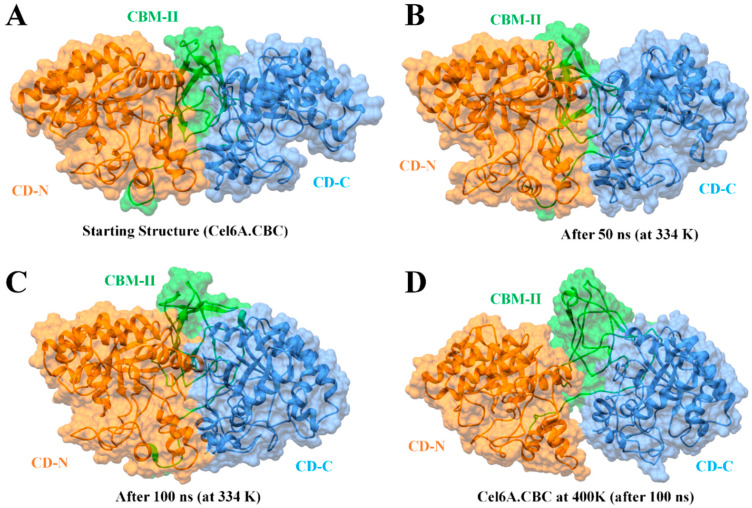
Molecular surface representation of cel6A.CBC construct with catalytic domains are colored as orange at N-terminal (CD-N) and cauliflower at C-terminal (CD-C), while CBM-II connecting the catalytic domains is colored green. (**A**) The initial conformation of cel6A.CBC before MD simulation. (**B**) The conformation of cel6A.CBC at an optimum temperature of 60.85 °C after 50 ns. (**C**) The conformation of cel6A.CBC at an optimum temperature of 60.85 °C after 100 ns. (**D**) The conformation of cel6A.CBC at an extreme temperature of 126.85 °C after 100 ns.

**Figure 7 biology-09-00214-f007:**
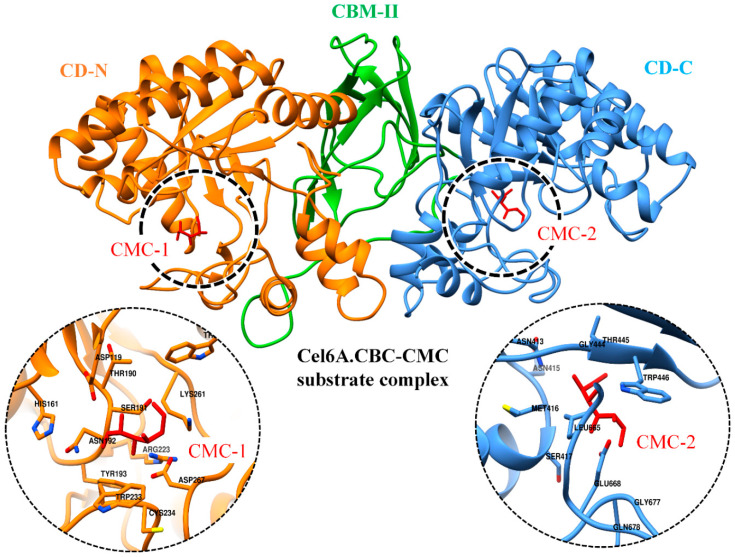
Docked conformation of CMC substrates (red sticks) as CMC1 and CMC2 with N and C-terminal catalytic domains of cel6A.CBC, respectively. The zoomed-in on left and right displaying the binding site residues in respective catalytic domains.

**Table 1 biology-09-00214-t001:** Expression levels of cel6A endoglucanase variants in *E. coli* and their relative activities against soluble and insoluble substrates.

Endoglucanase (cel6A) Variants	Cell Soluble Protein (mg l^−1^ OD_600_^−1^)	Expression Level of Variant in Soluble Cell Protein (%)	cel6A Yield (mg l^−1^ OD_600_^−1^)	Activity on 1% Carboxymethyl Cellulose (CMC)		Activity on Insoluble Substrates ^a^ (μmol Reducing Sugar μM^−1^ Enzyme)			
				Endoglu-Canase Activity ^a^ (U l^−1^ OD_600_^−1^)	Specific Activity (U mg^−1^ Enzyme)	Filter Paper (FP)	Bagasse	Regenerated Amorphous Cellulose (RAC)	Bacterial Microcrystalline Cellulose (BMCC)
cel6A.C	170 ± 8.7	45 ± 2.9	76.5 ± 3.9	320 ± 12.5	4.19 ± 0.05	0.7 ± 0.02	0.9 ± 0.06	1.3 ± 0.06	0.24 ± 0.03
cel6A.BC	168 ± 9.2	40 ± 3.5	67.2 ± 1.4	290 ± 10.2	4.32 ± 0.05	1.0 ± 0.05	1.4 ± 0.03	2.6 ± 0.12	0.28 ± 0.01
cel6A.BCB	190 ± 8.4	30 ± 2.1	57.0 ± 0.6	170 ± 9.9	2.98 ± 0.03	1.7 ± 0.09	2.2 ± 0.14	3.2 ± 0.18	0.60 ± 0.03
cel6A.CBC	185 ± 8.0	45 ± 3.0	83.3 ± 1.3	600 ± 14.8	7.20 ± 0.13	2.0 ± 0.11	3.4 ± 0.11	5.4 ± 0.25	0.90 ± 0.02
cel6A.CBCB	180 ± 10.3	20 ± 1.4	36.0 ± 0.3	250 ± 10.7	6.94 ± 0.31	1.5 ± 0.13	2.0 ± 0.10	1.7 ± 0.14	0.56 ± 0.09

**^a^** Endoglucanse activities were performed at 60 °C (pH 6.0) in triplicates.

**Table 2 biology-09-00214-t002:** Summary statistics of all-atom contacts and stereochemistry of cel6A.CB and cel6A.CBC after refinement through molecular dynamics (MD) simulations.

All-Atom Contacts	Summary Statistics	cel6A.CB (20 ns)		cel6A.CBC (20 ns)	
	Clash Score, All Atoms:	0.94		1.13	
Protein Geometry	Poor rotamers	8/407	1.96%	16/689	2.32%
	Favored rotamers	372/407	91.4%	632/689	91.72%
	Ramachandran outliers	7/407	1.71%	11/689	1.59%
	Ramachandran favored	370/407	90.9%	622/689	90.27%
	Ramachandran allowed	394/407	96.8%	674/689	97.9%
	MolProbity score	1.70		1.68	
	Cβ deviations >0.25Å	23	5.65%	51	7.4%
	Bad bonds:	30/2194	1.37%	62/5038	1.23%
	Bad angles:	74/3014	2.45%	105/6939	1.51%
Peptide Omegas	Cis Prolines:	0/20	0.00%	0/52	0.00%
	Cis nonProlines:	1	0.24%	3	0.42%

**Table 3 biology-09-00214-t003:** Molecular mechanics generalized Born surface area (MMGBSA) binding free energy results for cel6A.CB at standard temperature (26.85 °C) and cel6A.CBC at different temperatures.

Contributions	cel6A.CB (kcal/mol)	cel6A.CBC (kcal/mol)			
		26.85 °C	60.85 °C	74.85 °C	126.85 °C
ΔE_ele_	−5.55	−8.94	−7.85	−8.24	−2.15
ΔE_vdw_	−14.57	−23.87	−22.84	−20.26	−13.2
ΔE_MM_	−20.12	−32.81	−30.69	−28.5	−15.35
ΔG_p_	10.2	15.63	13.96	13.45	7.98
ΔG_np_	−2.09	−3.45	−4.82	−4.21	−1.62
ΔG_sol_	8.11	12.18	9.14	9.24	6.36
ΔG_tol_	−12.01	−20.63	−21.55	−19.26	−8.99

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
