# Peer review of "Enhanced Thermostability and Enzymatic Activity of cel6A Variants from Thermobifida fusca by Empirical Domain Engineering"

_biology, 2020, doi:10.3390/biology9080214_

Round 1
Reviewer 1 Report
General
There is no discussion about the characteristics of the new variants (temperature, thermal stability, and pH). It is not clearly stated which are the contributions of the study and how those characteristics are compared with the existing endoglucanases. Also, there is no comparison with the properties of the Cel6A wild type, nor experimentally or from literature.
Minor comments
Line 49-50: what do you mean with high temperatures? Specify the required temperature in the saccharification process
Line 62-63: is that all the available literature?
Line 64-66: please include major and latest references to support the sentence “…it have been extensively reported”
In line 75 and 199 it says “in vitro” what does it refer to?
Line 129: Which are the conditions for the ultra-sonication procedure?
In the section “Enzymatic activity assay” the conditions of the CMC-DNS assay are not specified. Which were the temperature, pH, buffer, CMC concentration, enzyme concentration used in the assay? How is the enzymatic unit (U) defined in your work? For the DNS assay, which was the utilized calibration curve? If the conditions were exactly the same as the reference [29] state so explicitly.
Line 151: specify the “regular intervals” at which the incubation was performed.
Line 154: Specify what was considered as 100% in the optimum activity.
Information from line 206 and 208-210 is already in the materials and methods, there is no need to put again in the results section.
Figure 3: specify the expected sizes for each construct.
Line 241: Cel6A.CBCB is described with 30 % in the manuscript and 20 % in Table 1, revise the data.
Line 257: Cel6A.CBCB is described with 1.7µM in the manuscript and 1.5 µM in Table 1, revise the data.
Figure 4 and Figure 6: Y axis is labeled as “endonuclease” and the research is about endoglucanases, revise.
Figure 5: The temperature symbols described in the legend of the figure does not match with the symbols in the graphs, revise. Also, axis in the figures are not uniform, this could lead to misinterpretation.

Reviewer 2 Report
It is very good paper and the readers could be attract about this investigation.
I can advise to improve the english in some part of the manuscript to give more lightness and fluidity to the text
Reviewer 3 Report
This is an interesting study which uses simple (but cleverly applied) techniques to generate and study variants of the Cel6A endoglucanase from Thermobifida fusca. I found the methods appropriate, the data appear sound, the conclusions are (mostly) appropriate to the results, and appropriate literature citation is present throughout. However, there are two overall points that I urge the authors to address before the manuscript is accepted for publication, to maximise the impact of their work.
- Comparison to native Cel6A
The significant weakness of this study is that no comparison is made to the native Cel6A. Appropriate data should be included throughout, but most importantly in Table 1 (activities), Figures 4-6 and Table 3, and comparison of the variants to the native enzyme should be made in the text. Appropriate comparison of Cel6A to the data in Figure 7 would also be useful (see also my comments on the molecular modelling section, below). This may simply be appropriately cited literature data, rather than new experimental measurements.
- Data presentation
There are a number of issues with the data presentation.
The first is relatively simple: Figures 4-6 appear to have smoothed linking lines between the data points, which is not appropriate. Lines of best fit to appropriate curves should be given instead.
The second is more serious: errors should be included throughout. There should be experimental errors on all columns in Table 1 except CelA6 yield and specific activity (which should have errors propagated from the other values). If numbers are included in the text (see my comments on lines 240-241 etc below), errors should also be given there. I suspect that the values for the activity of Cel6A.CBC and Cel6A.CBCB are within error of one another – if this is so then it alters the discussion somewhat. It’s nice to see error bars on Figures 4-6 but how are they calculated? They should be fully propagated from the two values (100% and experimental point) and the errors on those two values. Is this the case (especially given that no errors are given in Table 1)?
The third is a question of strengthening the discussion of the thermostability. It would be useful to see some discussion of Figure 7B in terms of the structure of Cel6A. Simply making this figure larger and putting a structural summary underneath (which regions are helix/sheet/loop/coil, where are the key active site residues?) would allow the reader to see how thermal fluctuations are affected by the local structure, and how they may affect the active site. It may also be useful to give the structure of Cel6A.CBC (and, ideally, Cel6A itself for comparison) coloured according to the RMSF values (a colour gradient from minimum to maximum RMSF, for example), to show how the global structure is affected. There would then be plenty of opportunity for further discussion. The conclusion says that the study “identified the factors responsible for thermostability”; inclusion of this structure-specific information would make this point even stronger.
I also have a number of suggestions for minor improvements to the text and figures:
There appear to be typing errors in the author addresses.
line 23: change “low hydrolytic activity and high instability” to “low stability and hydrolytic activity”
line 41: delete “biological”
line 42: delete “exclusively needed for pollutant fossil sources”
line 48: replace “vast” with “widespread”
line 58: delete “module of”
line 59: delete “in”
line 60: replace “helps” and “carries” with “help” and “carry”
line 121: the strain name is given incorrectly. Please provide the full name as given by the manufacturer. Most likely BL21-CodonPlus-RIPL or BL21-CodonPlus (DE3)-RIPL.
line 133: please give “DNS” in full here.
line 139: replace “lb.in2” with “lb.in–2”
line 195: delete “module of”
lines 240-241, 248-250, 256-257 and 259-260: numbers listed here are reproduced in Table 1. It is therefore not necessary to give every number in the text also. I suggest that you remove the lists of numbers from the text and instead just highlight trends and maxima (referenced to Table 1 to make it clear where the full data are).
lines 272-273: the text says that Cel6A.BC showed slightly lower activity at 55 °C – this does not appear to be true, based on Figure 4.
line 299: replace “3.0” with “5.0”.
line 305: once errors are included, it may not be appropriate to say that Cel6A.CBC had the highest activity.
Table 1: it would be useful to remind the reader that endoglucanase activity was measured at 60 ºC, as a baseline for later discussion. The pH should also be given. Footnotes a and b appear to be missing.
Figures 4-6: The format (text size, enzyme name) of the in-figure legends should be made consistent.
Figure 8: the in-figure caption for panel D is in a different format to the others. Please make it consistent.
Reviewer 4 Report
The biocatalytic conversion of plant biomass into bioethanol using cellulases is limited by low enzyme stability and catalytic efficiency under harsh industrial conditions i.e. high temperatures. In this work, Ali et al. performed domain engineering of an endoglucanase Cel6A from the thermophilic bacterium Thermobifida fusca to enhance both its catalytic efficiency and thermostability. This involved deleting, inserting and/or rearranging the enzyme’s catalytic domain and carbohydrate-binding domain. Each CelB6A variant’s activity (on soluble/insoluble substrates), pH optimum and thermostability was determined, with one variant Cel6A.CBC showing the good thermostability. Computer modelling was used to determine structural changes/influences on Cel6A.CBC’s activity at high temperatures, providing insights into its thermostability and potential future optimisation via enzyme engineering. Overall, work is of interest to enzyme engineers and industrial biotechnologists. Some of writing needs to be made clearer, some additional details, and a few cosmetic and structural changes (moving figures) are required.
Specific comments are as follows:
L51-53 This sentence needs to be written more clearly.
L54 Half a sentence to describe T. fusca is needed
L56-57 Cel6A is the ‘most’ efficient sugar producing enzyme from T. fusca? - or of all Cel6As? Make this sentence clearer.
L66 Grammar issue - make clearer
L67 Define MD here.
L73 change to “, we designed a group of Cel6A variants,”
L73-90 Too much detail for an intro - recommend this paragraph be written more concisely.
L94 ‘T4 ligase’ does not need to be in brackets.
Figure 1. This could go into the Supp. mats. under the primer listings.
To make things clearer for the reader, write ‘deleted, inserted, shuffled etc’. next to the I/II/III etc. The caption should also include more detail. Define ‘P’ (primer?)
L135 How was the DNS reaction halted? NaCO3 addition? … include.
L204 It would be easier for the reader to follow each variant/construct if Fig. S1 was included in the main text instead. Also, include ‘Truncated/inserted/Shuffled’ etc. next to each construct also in Fig S1.
L214 Considering the use of PCR, were the variants sequenced to verify their fidelity? Or was restriction enzyme digestion the only verification method used?
Figure 2. Could be moved to Supp. Mats.
Label with arrows on side of gel; (i.e. vector x) and (CelB6 genetic constructions)
L243 More detail on how enzyme concentrations were calculated for the enzymatic reactions is required here (perhaps a calculation in the methods section). Was a single batch used? Pooled batches? Did each batch have different concentrations?
Figure 3 Include name of each variant on the top of each sds-gel.
Write size (kDa) of each variant underneath each sds-gel
Include ‘…analysis of total soluble E.coli …’
Table 1 Define superscript a and b
These were performed in triplicate (see L145-146) – provide standard deviations.
Figure 3; Change X-axis heading from (degrees centigrade) to °C
L291 Given the article’s focus on thermostability, is it possible to also calculate the ½ life of each variant at each temp using existing data?
Figure 5 The Y-axis needs to be consistent for all graphs – they are not at present. This is critically important as it allows the reader to compare the data.
Figure 6 For consistency of terminology, correctly name each variant onto graph (currently abbreviated)
L306 ‘Figure 7, 8’ … should read ‘Figure 4 and 5’?
Figure 9/8 Incorrect figure numbers
